# Tetrel Bonding and Other Non-Covalent Interactions Assisted Supramolecular Aggregation in a New Pb(II) Complex of an Isonicotinohydrazide

**DOI:** 10.3390/molecules25184056

**Published:** 2020-09-04

**Authors:** Ghodrat Mahmoudi, Marjan Abedi, Simon E. Lawrence, Ennio Zangrando, Maria G. Babashkina, Axel Klein, Antonio Frontera, Damir A. Safin

**Affiliations:** 1Department of Chemistry, Faculty of Science, University of Maragheh, Maragheh P.O. Box 55181-83111, Iran; 2Department of Chemistry, Faculty of Science, University of Mohaghegh Ardabili, Ardabil P.O. Box 56199-11367, Iran; marjan_abedi2004@yahoo.com; 3School of Chemistry, Analytical and Biological Chemistry Research Facility, Synthesis and Solid State Pharmaceutical Centre, University College Cork, College Road, T12 K8AF Cork, Ireland; simon.lawrence@ucc.ie; 4Department of Chemical and Pharmaceutical Sciences, University of Trieste, Via L. Giorgieri 1, 34127 Trieste, Italy; ezangrando@units.it; 5Department für Chemie, Institut für Anorganische Chemie, Universität zu Köln, Greinstraße 6, D-50939 Köln, Germany; maria.babashkina@mail.ru; 6Department of Chemistry, Universitat de les Illes Balears, Crta de Valldemossa km 5.7, 7122 Palma de Mallorca Baleares, Spain; toni.frontera@uib.es; 7Institute of Chemistry, University of Tyumen, Volodarskogo Str. 6, 625003 Tyumen, Russia; 8West-Siberian Interregional Scientific and Educational Center, 625003 Tyumen, Russia; 9Innovation Center for Chemical and Pharmaceutical Technologies, Ural Federal University named after the First President of Russia B.N. Eltsin, Mira Str. 19, 620002 Ekaterinburg, Russia

**Keywords:** non-covalent interaction, tetrel bond, lead(II), isonicotinohydrazide, crystal structure, Hirshfeld surface analysis, DFT calculations

## Abstract

A new supramolecular Pb(II) complex **[PbL(NO_2_)]*_n_*** was synthesized from Pb(NO_3_)_2_, *N’*-(1-(pyridin-2-yl)ethylidene)isonicotinohydrazide (**HL**) and NaNO_2_. **[PbL(NO_2_)]*_n_*** is constructed from discrete [PbL(NO_2_)] units with an almost ideal N_2_O_3_ square pyramidal coordination environment around Pb(II). The ligand **L^−^** is coordinated through the 2-pyridyl N-atom, one aza N-atom, and the carbonyl O-atom. The nitrite ligand binds in a *κ*^2^-O,O coordination mode through both O-atoms. The Pb(II) center exhibits a hemidirected coordination geometry with a pronounced coordination gap, which allows a close approach of two additional N-atoms arising from the N=C(O) N-atom of an adjacent molecule and from the 4-pyridyl N-atom from the another adjacent molecule, yielding a N_4_O_3_ coordination, constructed from two Pb–N and three Pb–O covalent bonds, and two Pb⋯N tetrel bonds. Dimeric units in the structure of **[PbL(NO_2_)]*_n_*** are formed by the Pb⋯N=C(O) tetrel bonds and intermolecular electrostatically enforced π^+^⋯π^−^ stacking interactions between the 2- and 4-pyridyl rings and further stabilized by C–H⋯π intermolecular interactions, formed by one of the methyl H-atoms and the 4-pyridyl ring. These dimers are embedded in a 2D network representing a simplified uninodal 3-connected **fes** (Shubnikov plane net) topology defined by the point symbol (4∙8^2^). The Hirshfeld surface analysis of [PbL(NO_2_)] revealed that the intermolecular H⋯X (X = H, C, N, O) contacts occupy an overwhelming majority of the molecular surface of the [PbL(NO_2_)] coordination unit. Furthermore, the structure is characterized by intermolecular C⋯C and C⋯N interactions, corresponding to the intermolecular π⋯π stacking interactions. Notably, intermolecular Pb⋯N and, most interestingly, Pb⋯H interactions are remarkable contributors to the molecular surface of [PbL(NO_2_)]. While the former contacts are due to the Pb⋯N tetrel bonds, the latter contacts are mainly due to the interaction with the methyl H-atoms in the π⋯π stacked [PbL(NO_2_)] molecules. Molecular electrostatic potential (MEP) surface calculations showed marked electrostatic contributions to both the Pb⋯N tetrel bonds and the dimer forming π^+^⋯π^−^ stacking interactions. Quantum theory of atoms in molecules (QTAIM) analyses underlined the tetrel bonding character of the Pb⋯N interactions. The manifold non-covalent interactions found in this supramolecular assembly are the result of the proper combination of the polyfunctional multidentate pyridine-hydrazide ligand and the small nitrito auxiliary ligand.

## 1. Introduction

Non-covalent interactions were first recognized by J. D. van der Waals in his doctoral thesis about one and a half century ago [1]. The most prominent example for the crucial role of non-covalent interactions is probably the double helix structure of DNA [2,3]. Different types of non-covalent interactions, such as hydrogen bonding, π⋯π interaction, halogen bonding, chalcogen bonding, pnictogen bonding, tetrel bonding, (an)agostic bonding, and cation/anion⋯π interaction, can be distinguished, and their role in all areas of molecular chemistry and biology is eminent [3,4,5,6,7,8,9,10,11,12,13,14,15,16,17,18,19,20,21,22,23]. Amongst them, non-covalent π⋯π interactions, also called π-stacking, found between aromatic systems, are of great interest due to their broad applications [4,5,6,7,8,9,10,11,13,14,15,16,18]. Moreover, «stacking interactions» can also be addressed to aliphatic systems [14]. Notably, benzene and cyclohexane interact more efficiently (−3.01 kcal/mol) [18] than benzene (−2.758 kcal/mol) [14] and cyclohexane (−2.62 kcal/mol) [7] dimers.

On the other hand, the coordination chemistry of Pb(II) gains particular interest from the large ionic radius of this heavy p-block metal ion, its rich variety of coordination numbers from 2 up to 10, and the peculiar feature of the 6s^2^ lone-pair in the coordination sphere of its complexes [17,19,22,24,25,26,27,28,29,30]. This lone-pair can be stereochemically active, which is called hemidirectional or non-active, for which the term “holodirectional” was coined ([Fig molecules-25-04056-ch001]) [19,24,25,26,27,28,29,30]. The background is the observation that the electron pair on the Pb(II) atom is either stereochemically irrelevant (holodirected) or demanding space (hemidirected) [17,19].

The hemidirected coordination allows Pb(II) to participate in the formation of tetrel bonds, which are of great importance for the resulting topology of Pb(II) coordination compounds [17,19,22,24,25,26,27,28,29,30,31,32,33,34,35,36,37,38,39,40,41,42]. Although the occurrence of hemidirected or holodirected bonding cannot be reliably predicted [19,28,29,30,31,40,41], recent work using polyfunctional multidentate ligands, such as carboxylates of 2-thiols or heterocycles [19,24,25,26,27,28], or the per se polyfunctional thiothiosemicarbazones, hydrazones, or Schiff base ligands of the salen or salan type [29,30,31,32,33,34,35,36,37,38,39,40,41,42] allowed formation of tetrel bonds. We also have recently contributed to this and have found the polyfunctional 1-(pyridin-2-yl)ethylideneisonicotinohydrazide ligand (**HL**, Scheme 1) extremely versatile to build up supramolecular Pb(II) complexes or coordination polymers with rich non-covalent interactions, including the reliable formation of tetrel bonding [32,38,42]. Particularly, we have studied the heteroleptic complexes and coordination compounds of the type **[PbL(SCN)]*_n_*** and **[Pb(HL)(NCS)_2_]*_n_*** [32], and **[Pb(HL)Cl_2_]*_n_*** [38]. From the isomeric 1-(pyridin-2-yl)ethylidenenicotinohydrazide ligand (**HL’**, Scheme 1) compounds of the type **[Pb_2_(HL’)_2_(NO_3_)_2_(NCS)_2_]**, **[PbL’(OAc)]_2_**, **{[Pb(HL’)(OAc)]ClO_4_}*_n_***, **{[PbL’]ClO_4_}*_n_*·nH_2_O**, **[PbL(N_3_)]*_n_*** and **[PbL’(NO_2_)]*_n_*** [37], and **[Pb(HL’)(NO_3_)_2_]*_n_*** and **[PbL’(CH_3_O)]*_n_*** [39] have been studied. We recently reported also the homoleptic complex **[Pb(L’’)_2_]** containing the *N*’-(4-hydroxybenzylidene)isonicotinohydrazide ligand [42].

In view of the interesting structure of the previously described nitrito nicotinohydrazide compound **[Pd(L’(NO_2_)]*_n_*** [37], we reacted a mixture of Pb(NO_3_)_2_ and 1-(pyridin-2-yl)ethylideneisonicotinohydrazide (**HL**) with NaNO_2_ and received the corresponding isonicotino isomer **[PbL(NO_2_)]*_n_***, which turned out to have a completely different structure compared with **[PbL’(NO_2_)]*_n_*** (Scheme 1). The nature and energetic features of the two Pb∙∙∙N tetrel bonds were studied by density functional theory (DFT) and molecular electrostatic potential (MEP) calculations, which demonstrate the presence of a σ-hole at the Pb(II) ion.

## 2. Results and Discussion

The reaction of a mixture of Pb(NO_3_)_2_ and **HL** with NaNO_2_ in MeOH led to a new supramolecular heteroleptic coordination compound **[PbL(NO_2_)]*_n_*** (Scheme 1). The compound was isolated as a crystalline air- and moisture-stable solid that was characterized through elemental analysis, FTIR spectroscopy, single-crystal X-ray diffraction, and Hirshfeld surface analysis.

The FTIR spectrum of **[PbL(NO_2_)]*_n_*** contains characteristic bands for the C=O and C=N bonds at 1632 and 1590 cm^−1^, respectively. The methyl C–H fragments showed resonances at 2921 cm^−1^.

Compound **[PbL(NO_2_)]*_n_*** crystallized in the monoclinic space group *P*2_1_/*n*, with one coordination unit [PbL(NO_2_)] in the asymmetric unit. The deprotonated **L**^−^ and the nitrito NO_2_^−^ ligands were covalently bound to the metal center, yielding an almost ideal N_2_O_3_ square pyramidal coordination environment around the Pb(II) atom (Figure 1), as evidenced from the so-called *τ*_5_-descriptor of about 0.08 (Table 1) [43]. The ligand **L**^−^ is tridentately coordinated through the 2-pyridyl and the aza N-atoms and the carbonyl O-atom. The nitrite ligand binds in a *κ*^2^-O,O coordination mode through both O-atoms to Pb(II).

This structure is completely different from that of the previously reported nicotinohydrazide isomer **[PbL’(NO_2_)]*_n_*** [37]. In this coordination polymer, Pb(II) shows a distorted, non-octahedral six-fold coordination in which the pending 3-pyridyl N-atom bridges between two metals (Scheme 1 and Table 1).

The Pb–N bond lengths in **[PbL(NO_2_)]*_n_*** are 2.462(4) and 2.583(4) Å, with Pb–N(N) < Pb–N_2-Py_. The Pb–O distance with the carbonyl oxygen atom is shorter and of 2.387(3) Å. These values are in line with data from related hydrazide Pb(II) complexes [32,35,37,38,39,42]. The same bonds with the nitrite oxygen atoms differ significantly. Particularly, while one of the bonds has the same length as that formed with the carbonyl O atom, the second bond with 2.768(4) Å is the longest within the covalent bonds formed by Pb(II) in the structure of **[PbL(NO_2_)]*_n_***.

The covalent bond distances around the Pb(II) atom in **[PbL(NO_2_)]*_n_*** are very similar to the nicotinohydrazide isomer **[PbL’(NO_2_)]*_n_*** [37], and, in both cases, the corresponding organic ligand is almost completely planar, as can be seen from the torsion angles of only about 3° between the mean planes formed by the pyridyl rings (Table 1).

The covalent bonds in the central coordination unit are markedly concentrated on one hemisphere of the coordination environment (Figure 1), representing a hemidirected coordination geometry with a pronounced coordination gap, due to the 6s^2^ lone pair. This allows for two additional N-atoms from adjacent molecules, the aza and the 4-pyridyl N-atoms, to approach the Pb(II) and form two Pb⋯N tetrel bonds (Figure 2). The Pb⋯N=C(O) tetrel bond (3.235(4) Å) is significantly longer than the sum of the covalent radii (2.17 Å) and shorter than the sum of van der Waals radii (3.57 Å), thus supporting its non-covalent nature. It leads to the formation of centrosymmetric dimers [PbL(NO_2_)]_2_ (Figure 2), which are further reinforced by bilateral π⋯π stacking interactions, formed between the 2- and 4-pyridyl rings (Figure 2 and Table 2). These dimers are interlinked through bilateral C–H⋯π intermolecular interactions, formed by one of the methyl H-atoms and the 4-pyridyl ring (Figure 2 and Table 3). The slightly shorter Pb⋯N_4-Py_ tetrel bond (3.039(5) Å) leads to a supramolecular aggregation of the dimers into a 2D layer structure (Figure 3). From a topological perspective, this 2D metal–organic layer in **[PbL(NO_2_)]*_n_*** is assembled from the 3-connected Pb(II) nodes and 3-connected **L** linkers and can be classified as a uninodal 3-connected **fes** (Shubnikov plane net) [44] topology defined by the point symbol (4∙8^2^). Thus, the overall topology of **[PbL(NO_2_)]*_n_*** is a supramolecular 3D framework, formed by multiple non-covalent interactions, such as Pb⋯N tetrel bonds, π⋯π stacking, and C–H⋯π interactions.

This stands in complete contrast to the previously reported nicotinohydrazide isomer **[PbL’(NO_2_)]*_n_***, in which a similar Pb⋯N=C(O) tetrel bond and a very dissimilar Pb⋯ONO tetrel bond lead to dimers, which are interconnected through the Pb‒N_3-Py_ coordinative bond, thus forming a zigzag polymeric structure and only one remarkable π⋯π stacking interaction with a centroid–centroid distance of 3.615(3) Å [37].

To further examine the intermolecular interactions in the crystal of **[PbL(NO_2_)]*_n_***, we used a Hirshfeld surface analysis [45]. This analysis provides an excellent overview over all intermolecular interactions originating from a species, regardless of its nature. Together with the analysis of the nature of these interactions (XRD data, molecular electrostatic potential (MEP), binding energies from DFT, and quantum theory of atoms in molecules (QTAIM) calculations), it gives a complete picture, whose forces contribute to the molecular and crystal structure. The 2D fingerprint plots [46] of the basic coordination unit [PbL(NO_2_)] were generated, using CrystalExplorer 3.1 [47], to visualize proportions of the intermolecular interactions. Furthermore, we calculated the enrichment ratios (*E*) [48] of the intermolecular contacts to estimate the probability of two chemical species to be in contact.

The Hirshfeld surface analysis showed that the intermolecular H∙∙∙X (X = H, C, N, O) contacts occupy an overwhelming majority of the molecular surface of the basic coordination unit [PbL(NO_2_)] (Figure 4 and Table 4). The shortest H∙∙∙H contacts are shown in the corresponding 2D fingerprint plot at *d*_e_ + *d*_i_ ≈ 2.3 Å, and a clear splitting of the H∙∙∙H fingerprint was found (Figure 4), which is due to the shortest contact being between three atoms, rather than for a direct two-atom contact [45]. The H∙∙∙C contacts in the corresponding 2D plot of [PbL(NO_2_)] were found in the form of «wings», with the shortest being *d*_e_ + *d*_i_ ≈ 2.8 Å (Figure 4). This is characteristic for C–H⋯π type of interactions [45]. The H∙∙∙N and H∙∙∙O contacts are shown in the corresponding 2D fingerprint plots as two «horns», with the shortest being *d*_e_ + *d*_i_ ≈ 2.4–2.5 Å (Figure 4).

Furthermore, the structure of [PbL(NO_2_)] is also characterized through intermolecular C∙∙∙C and C∙∙∙N interactions (Figure 4) observed as the area at *d*_e_ = *d*_i_ ≈ 1.7–2.0 Å. They correspond to intermolecular π⋯π stacking interactions between the pyridyl rings. Importantly, intermolecular Pb∙∙∙N and, most interestingly, Pb∙∙∙H interactions are also remarkable contributors into the molecular surface of [PbL(NO_2_)]. The former contacts are shown in the corresponding 2D fingerprint plot as two sharp «spikes», with the shortest being *d*_e_ + *d*_i_ ≈ 3.0 Å (Figure 4), and correspond to the Pb⋯N tetrel bonds. The latter contacts are mainly due to the interaction with the methyl H-atoms arising from an adjacent π⋯π stacked [PbL(NO_2_)] unit.

The 2D plot of [PbL(NO_2_)] exhibits points at large *d*_e_ and *d*_i_ (Figure 4), which are similar to those observed in the 2D plots of benzene [45] and have been observed in similar compounds [49,50,51,52,53,54]. They correspond to regions without any close contacts to adjacent molecules.

The favorable H∙∙∙X (X = H, N, O, Pb) contacts in the structure of **[PbL(NO_2_)]*_n_*** show enrichment ratios *E*_HH/HN/HO/PbH_ close to or even higher than unity (Table 4). The H∙∙∙C, C∙∙∙N, and N∙∙∙N contacts are much less favored, and they are consistent with smaller enrichment ratios of *E*_HC/CN/NN_ = 0.82–0.87. This is explained by the presence of the highly probable C∙∙∙C (*E*_CC_ = 2.45) and Pb∙∙∙N (*E*_PbN_ = 3.46) contacts in the structure of the compound. Remaining contacts are discouraged with enrichment ratios ranging from 0.00 to 0.48 (Table 4).

As the Pb⋯N=C(O) tetrel bonding and the π⋯π stacking interactions are the dominating forces for the formation of the dimers in the solid state of **[PbL(NO_2_)]***_n_* (Figure 2 and Figure 3), we further analyzed the donor–acceptor properties of the basic coordination unit [PbL(NO_2_)] through DFT calculations.

First, we computed the molecular electrostatic potential (MEP) surface and found the highest positive values at the Pb(II) atom (+26.7 kcal/mol) representing the σ-hole (Figure 5). The most negative part is located at the 4-pyridyl N-atom (−40.8 kcal/mol) of the ligand **L^−^**, which is more electron-richer than the nitrito N- and O-atoms (−31.4, −33.3, and −37.6 kcal/mol). Consequently, the Pb⋯N_4-Py_ tetrel bonding interaction is the most favored contact from an electrostatic point of view in line with the observed structure, *d*(Pb⋯N_4-Py_) = 3.039(5) Å (Table 1). The second Pb⋯N=C(O) tetrel bond is much weaker (3.235(4) Å) and consistent with a significantly less negative value at the N=C(O) N-atom (−21.3 kcal/mol). From these MEP calculations, a relatively high probability of a Pb⋯ONO tetrel bond can be concluded. However, no such bond was observed in the structure of **[PbL(NO_2_)]***_n_*. In contrast to this, in the nicotinohydrazide isomer **[PbL’(NO_2_)]*_n_***, such a Pb⋯ONO tetrel bond was observed, although not very dominant and rather long (3.299(4) Å, Table 1).

An interesting feature of the surface is that the MEP values over the coordinated and non-coordinated pyridine ring centers represent opposite signs (Figure 5). As a consequence, they are well suited to form strong, electrostatically enforced π^+^⋯π^–^ interactions [6] between the coordinated and non-coordinated pyridine rings of the **L^‒^** ligand, in line with the dimer-promoting stacking interactions found in **[PbL(NO_2_)]***_n_* (Figure 2 and Table 2).

The DFT-calculated binding energy of these antiparallel stacked dimers composed of electrostatically enforced π^+^⋯π^–^ and Pb⋯N=C(O) tetrel interactions (Figure 6) is quite large (Δ*E*_1_ = −24.3 kcal/mol), in agreement with the MEP surface analysis. In contrast to this, the calculated interaction energy of the Pb⋯N_4-Py_ tetrel bond is moderately strong (Δ*E*_2_ = −10.2 kcal/mol) and comparable to recently reported values [32,35]. When looking at the bond (red spheres), ring (yellow spheres), and cage (blue spheres) critical points (CPs), we found that the π^+^⋯π^−^ interaction is characterized by two bond CPs that interconnect two atoms of each pyridine ring (Figure 6). The π⋯π stacking is further characterized by additional CPs that interconnect the chelate rings and more ring and cage CPs. Moreover, the two Pb⋯N=C(O) tetrel bonds are characterized by bond CPs and bond paths (Figure 6). The Pb⋯N_4-Py_ tetrel bond contributes, together with a C‒H⋯O interaction, to the overall binding energy of entire assembly (Figure 6). The energy associated to this C–H⋯O interaction was estimated by using the kinetic energy density predictor (*E* = 0.5 × Vr) to 1.3 kcal/mol. The Pb⋯N_4-Py_ tetrel bond interaction is thus significantly stronger in keeping with the quite short experimentally observed distance of 3.039(5) Å. Thus, this tetrel bond dominates this type of intermolecular interaction and drives the formation of the uninodal 3-connected **fes** (Shubnikov plane net) topology.

## 3. Materials and Methods

### 3.1. Reagents

All reactants were used as received, without further purification. The ligand **HL** was synthesized by following the available protocol [37].

### 3.2. Instrumentation

FTIR spectra were recorded on a Bruker Tensor 27 FTIR spectrometer (Bruker, Ettlingen, Germany). Microanalyses were performed, using a Heraeus CHN-O-Rapid analyzer (Heraeus, Hanau, Germany).

### 3.3. Synthesis of [PbL(NO_2_)]_n_

A solution of **HL** (0.024 g, 0.1 mmol) in MeOH (10 mL) was added dropwise to a solution of Pb(NO_3_)_2_ (0.033 g, 0.1 mmol), in the same solvent (10 mL). To this mixture, a solution of NaNO_2_ (0.014 g, 0.2 mmol) in MeOH (5 mL) was added slowly under stirring. The resulting mixture was stirred at room temperature for 30 min and was left undisturbed for slow evaporation. After about 4 days, yellow X-ray suitable single crystals were formed. Crystals were isolated by filtration. Yield: 0.038 g (78% based on Pb(NO_3_)_2_). FTIR, ν: 2921 (CH), 1632(C=O), 1590 (C=N), 1269 (NO_2_) cm^−1^. *Anal.* Calc. for C_13_H_11_N_5_O_3_Pb (492.46): C 31.71, H 2.25, and N 14.22; found: C 31.80, H 2.28, and N 14.18%.

### 3.4. Single-Crystal X-Ray Diffraction of [PbL(NO_2_)]_n_

The data were collected with a Bruker AXS SMART APEX2 CCD (Bruker, Ettlingen, Germany) diffractometer, operating at 179(2) K and using graphite monochromated Mo-Kα radiation (λ = 0.71073 Å). Cell refinement, indexing, and scaling of the datasets were performed, using the program Bruker Smart Apex and Saint packages [55]. The structure was solved by direct methods and refined by full-matrix least-squares on *F*^2^ with anisotropic displacement parameters for all non-hydrogen atoms, using the program SHELXL [56]. The H atoms were included as riding contributions with fixed isotropic displacement parameters in idealized positions. All the calculations were carried out by using the WinGX System, V2013.3 [57]. Crystal structure determination and refinement data are given in Table 5. CCDC-2014301 contains the Appendix A for this paper (Appendix A). These data can be obtained free of charge via http://www.ccdc.cam.ac.uk/conts/retrieving.html (or from the CCDC, 12 Union Road, Cambridge CB2 1EZ, UK; Fax: +44 1223 336033; E-mail: deposit@ccdc.cam.ac.uk).

### 3.5. DFT Calculations

Gaussian-16 package [58] was used to perform the density functional theory calculations reported herein. In particular, the PBE1PBE-D3 functional [59,60], in combination with the def2-TZVP basis set [61,62], was employed for computing the binding energies and the molecular electrostatic potential (MEP) surfaces. The crystallographic coordinates were used for the calculations. The 0.001 a.u. isosurface was used for mapping the MEP onto the van der Waals surface. The QTAIM analysis [63] was carried out at the same level of theory and using the program AIMAll [64].

## 4. Conclusions

A new supramolecular Pb(II) coordination compound **[PbL(NO_2_)]*_n_*** was synthesized through treating a mixture of Pb(NO_3_)_2_ and *N’*-(1-(pyridin-2-yl)ethylidene)isonicotinohydrazide (**HL**) with NaNO_2_. The yellow material is composed of discrete mononuclear heteroleptic [PbL(NO_2_)] coordination units containing the tridentate N^N^O binding deprotonated ligand **L^−^** and a *κ^2^*-O,O binding nitrito NO_2_**^−^** ligand. Pb(II) exhibits a hemidirected coordination geometry with a pronounced coordination gap. This allows two Pb⋯N=C(O) tetrel bonds of the uncoordinated N-atoms of two symmetry related [PbL(NO_2_)] molecules forming pronounced dimers in the crystal and Pb⋯N_4-Py_ tetrel bonds to the pending 4-pyridyl group forming a 2D network representing a simplified uninodal 3-connected **fes** (Shubnikov plane net) topology defined by the point symbol (4∙8^2^). Thus, Pb(II) is in a N_4_O_3_ coordination environment, formed by two covalent Pb–N, three covalent Pb–O bonds, and two Pb⋯N tetrel bonds. The Pb⋯N tetrel bonds in the crystal packing of **[PbL(NO_2_)]*_n_*** are further reinforced by bilateral intermolecular π⋯π stacking interactions, formed between the 2- and 4-p.

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
