# Peer review of "Tetrel Bonding and Other Non-Covalent Interactions Assisted Supramolecular Aggregation in a New Pb(II) Complex of an Isonicotinohydrazide"

_molecules, 2020, doi:10.3390/molecules25184056_

Round 1

Reviewer 1 Report

The paper of Klein and et. al reports the crystallographic and computational studies of a single molecule composed by [PbL(NO2)]n, which was synthesized from Pb(NO3)2, N’-(1-(pyridin-2-yl)ethylidene)isonicotinohydrazide (HL) and NaNO2.

The paper is well-written and all the data and results are described clearly.

All the characterization are performed very accurately.

The results are supported by the presence of a very detailed crystallographic and computational analysis.

The computational studies show each single contribute of different non-covalent interactions detected in the crystal and all approaches (MEP, BCP, crystalexplorer) highlight the role of Pb…N interaction.

This work shows how a TtB synthon can work in a cooperative way with other non-covalent interactions  to drive the assembly of Pb-molecule in solid state. The results reported in this paper are very useful for the crystal engineering community and add an import piece of information in the crystal engineering tools box: the PB…N interaction is a reliable TtB synthon.

In my opinion the paper can be published in Molecules in the present form.

Author Response

We thank the reviewer for considering our manuscript to be suitable for publication in Molecules.

Reviewer 2 Report

This manuscript entitled "Tetrel bonding and Other Non-covalent Interactions Assisted Supramolecular Aggregation in a New Pb(II) Complex of an Isonicotinyohydrazide" reports structural studies of Pb(II) coordinated to pyridine and hydrazide ligands. The X-ray crystallography results show the complex is stabilized by a number of non-covalent interactions including π-π stacking, Pb-N tetrel bonds. This interplay of interactions are complemented with MEP surface calculations as well as QTAIM analyses. I recommend publication of this manuscript with minor revisions.

Comments:

Abstract - too long and wordy. It should be more concise.

Introduction

Page 1 Line 60-63: This sentence needs to be re-arranged. Awkward sentence.

Page 1 Line 87: delete "Very" and consider "We recently reported"

Page 4 Line 118: This is completely different "from" not "to"

Page 4 Line 121-124: Consider rearranging the sentence.

Page 5 Line 129: "very small" small compared to what?

Page 5 Line 146: "tetrel bond" What is the sum of van der Waals radii of Pb and N? Can authors comment on the length of these tetrel bond? How much longer than sum of van der Waals radii and how much shorter than typical Pb-N covalent bond?

Page 6 Line 177: Why did authors choose to use Hirshfeld surface analysis? What are the advantages/disadvantages of using this particular method for studying sigma-hole interactions?

Page 8 Line 214: Consider "we further analyzed"

Page 9 Lines 244-247: Awkward sentence. Please rearrange.

Figure 2, 3, 4 and 6: Quality of the images need to be improved. They look pixelated when printed on paper.

Figure 4 and 5: The caption lacks details of the image. Please mention what each color means for both Figures, what are the axes (Figure 4), what is the electron density representation in Figure 5?

Author Response

We thank the reviewer for the valuable comments.

Comment: Abstract - too long and wordy. It should be more concise.

Response: The abstract was slightly modified. We believe further shortening leads to a loss of important information.

Comment: Introduction, Page 1 Line 60-63: This sentence needs to be re-arranged. Awkward sentence.

Response: We agree that this is a lot of information in one sentence and we have tried to rectify it for better readability

Comment: Page 1 Line 87: delete "Very" and consider "We recently reported"

Response: Yes, corrected.

Comment: Page 4 Line 118: This is completely different "from" not "to"

Response: Thank you, corrected.

Comment: Page 4 Line 121-124: Consider rearranging the sentence.

Response: We have revised the complete sequence, making shorter sentences.

Comment: Page 5 Line 129: "very small" small compared to what?

Response: We have revised this part to make it clearer.

Comment: Page 5 Line 146: "tetrel bond" What is the sum of van der Waals radii of Pb and N? Can authors comment on the length of these tetrel bond? How much longer than sum of van der Waals radii and how much shorter than typical Pb-N covalent bond?

Response: This information has been added in the manuscript and commented.

Comment: Page 6 Line 177: Why did authors choose to use Hirshfeld surface analysis? What are the advantages/disadvantages of using this particular method for studying sigma-hole interactions?

Response: We have added the following:

To further examined the intermolecular interactions in the crystal of [PbL(NO2)]n we used a Hirshfeld surface analysis [45]. This analysis provides an excellent overview over all intermolecular interactions originating from a species regardless of their nature. Together with the analysis of the nature of these interactions (XRD data, molecular electrostatic potential (MEP), binding energies from DFT, and quantum theory of atoms in molecules (QTAIM) calculations) it gives a complete picture which forces contribute to the molecular and crystal structure.

Comment: Page 8 Line 214: Consider "we further analyzed"

Response: Corrected.

Comment: Page 9 Lines 244-247: Awkward sentence. Please rearrange.

Response: We have rearranged this paragraph to make it clearer.

Comment: Figure 2, 3, 4 and 6: Quality of the images need to be improved. They look pixelated when printed on paper.

Response: All Figures were reproduced at >1200 dpi quality.

Comment: Figure 4 and 5: The caption lacks details of the image. Please mention what each color means for both Figures, what are the axes (Figure 4), what is the electron density representation in Figure 5?

Response: We have added some more information to the captions.

Reviewer 3 Report

  This manuscript continues along the lines the authors have established in their past work, studying a somewhat different molecular system.  The emphasis here is on the intermolecular contacts, honing in on particular noncovalent interactions, and trying to assess their importance.  The bulk of the paper concerns analysis of the crystal structure, and precisely defining the intermolecular contacts.  The results are presented in a digestible manner, with useful figures and tables.  The last section describes quantum calculations to further analyze these contacts from another perspective.  The authors analyze their results competently so as to offer some quantitative estimates as to the various bonds.  The energetic and AIM analyses are useful in this regard and the results are described in an understandable manner.  The paper represents a step forward in understanding Pb and some of its noncovalent bonds, so publication can be recommended.  The only suggestion would be for the authors to supply the values of the densities at the bond critical points so that a reader can make a detailed comparison of the various interactions.

Author Response

Thank you for the suggestion. We have added these values to the figure.

Reviewer 4 Report

The work of Mahnoudi et al., report  preparation,  crystal structure and analysis of intermolecular bonding in the new Pb(II)  complex of  isonicotinohydrazide. This compound represents another entry in a series of the similar complexes reported by the authors earlier. The experimental and computational work is sound, and the structure comprise an interesting example of the tetrel bonding and deserves publication.  The moderate English language editing of the manuscript is needed, but otherwise it requires minor revisions. In particular:

  • The abstract seems way too long and should be shortened.
  • Introduction starts with very general statements, but provide too little specific background and objectives.
  • Bond angles from Table 1 can be relegated to SI.
  • Rings I and J in Tables 2 and 3 should be explained
  • Since Table  3 contains only 1 line, the data could be presented in the text.  Also, crystallographic data for one compound can be presented as one paragraph.

Author Response

The moderate English language editing of the manuscript is needed, but otherwise it requires minor revisions. In particular:

The abstract seems way too long and should be shortened

Response: The abstract was slightly modified. We believe further shortening leads to a loss of important information. The English was revised.

Comment: Introduction starts with very general statements, but provide too little specific background and objectives

Response: We have added Chart 1 and more information to the introduction.

Comment: Bond angles from Table 1 can be relegated to SI.

Response: For the discussion we need both bond lengths and angles. Moreover, there is no SI so far and we refrained from creating one only for the purpose of hosting the bond angles.

Comment: Rings I and J in Tables 2 and 3 should be explained

Response: The rings “2-Py” and “4-Py” are now explained in the tables.

Comment: Since Table 3 contains only 1 line, the data could be presented in the text. Also, crystallographic data for one compound can be presented as one paragraph.

Response: Basically, the referee is right. But we prefer to keep it as a Table since we believe this way of presentation is more eye-catching for readers and for data analysis.

Reviewer 5 Report

This manuscript represents comprehensive research on lead(II) complex forming tetrel bonds in the crystal structure. The authors used modern quantum chemical calculation methods and clearly described the results of the research. But, in my opinion, the manuscript lacks a comparison of the obtained results with the known literature data. It is not clear from the text whether the results unique or just typical for the lead(II) metal center. I recommend to improve the manuscript in this way before resubmission.

Author Response

We have revised the manuscript carefully to meet the concerns of the reviewer. The introduction now gives more background. The comparison between the new complex and the previously reported derivative starts with Scheme 1 and continues throughout the manuscript. The XRD data is compared to similar complexes. Unfortunately, for the very detailed analysis such as Hirshfeld, MEP and QTAIM only a few comparisons could be drawn. This also underlines the importance and novelty of the here presented results.